# OTCOP: LEARNING OPTIMAL TRANSPORT MAP VIA CONSTRAINT OPTIMIZATION

## ABSTRACT

The approximation power of the neural network makes it an ideal tool to learn optimal transport maps. However, existing methods are mostly based on the Kantorovich duality and require regularizations and/or special network structures. In this paper, we propose a direct constraint optimization algorithm for the computation of optimal transport maps based on the Monge formulation. We solve this constraint optimization problem by using three different methods: the Langrangian multiplier method, the augmented Lagrangian method, and the alternating direction method of multipliers (ADMM). We demonstrate a significant accuracy of learned optimal transport maps on high dimensional benchmarks. Moreover, we show that our methods reduce the regularization effects and accurately learn the target distributions at a lower transport cost.

## 1 INTRODUCTION

There has been a great interest in applying modern machine learning techniques for finding optimal transport maps between two distributions. Different from traditional computational methods that solve PDEs for optimal transport maps (Benamou & Brenier (2000); Angenent et al. (2003); Li et al. (2018)), modern machine learning techniques aim to solve the problem directly by optimizations. The Sinkhorn Distance method Cuturi (2013); Peyré et al. (2019), the regularized OT dual Seguy et al. (2017) have been used to find large scale optimal transport maps between discrete probability distributions and have been used to train generative networks Genevay et al. (2018); Sanjabi et al. (2018). A geometric treatment is provided in Gu et al. (2013). The Input Convex Neural Network (ICNN) is used to construct a convex Brenier potential for finding optimal transport maps Makkuva et al. (2020) between continuous distributions and is recently used in population dynamics Bunne et al. (2022), which combines the ICNN and Sinkhorn distance methods Amos et al. (2022). Despite these successes, most methods are based on the duality formulation and avoids the direct treatment on the Monge problem.

In this paper, we focus on the direct solution of the Monge problem. The Monge problem (Monge (1781)) directly seeks to identify the optimal transport maps and is a nonlinear constraint optimization problem. The major difficulty in solving the problem numerically is that it is nonlinear and includes a constraint that the push-forward distribution is equal to the target distribution, which is difficult to implement. Therefore, most optimal transport algorithms avoid directly solving the Monge problem but use the Kantorovich duality (Kantorovich (1942)), for which the objective function is linear and the transport map is obtained by taking the gradient of the Brenier potential for the quadratic cost. However, these two problems are not always identical (Villani (2009)) and it is desirable to find a direct approach for the Monge problem.

The Monge problem has been solved numerically using optimization based methods with polynomial approximations. For example, a Lagrangian penalty method was used to find optimal transport maps approximated by polynomials for Bayesian inference El Moselhy & Marzouk (2012) and space discretization was used in Haber et al. (2010) to calculate the Jacobian matrix of the transport maps and transferred the optimization to finite dimensional spaces. However, their approaches are limited to low dimensions as number of grids expands exponentially as dimensions become large. Considering the success of deep neural networks in approximating high dimensional data, the integration of classical constraint optimization methods and neural networks holds a promise.

One successful application of the optimal transport theory to deep learning is the Wasserstein Generative Adversarial Network (WGAN) Arjovsky et al. (2017). However, WGAN only use the optimal transport distance as a loss function and does not target at finding the optimal transport maps. It is desirable to study whether it is possible to lower the transport cost of the map learned by WGAN or other networks using the algorithm for finding optimal transport maps.

This paper presents a new approach for finding optimal transport maps between two continuous distributions. We make the following contributions:

- We integrate three constraint optimization algorithms including the Standard Lagrangian (SL), the Augmented Lagrangian method (AL) and the Alternating Direction Method of Multipliers (ADMM) with neural networks to solve the Monge problem of optimal transport with provable guarantees (Theorem 1-3).

- We show that our method is able to find an accurate optimal transport map between Gaussian distributions, both theoretically (Theorem 2) and experimentally. Moreover, we apply our method to WGAN and show that our method can find a generative map with lower transport cost while not sacrificing the quality of outputs.

- We compare the three algorithms and find the SL algorithm introduces errors but is simple and easy to implement, while AL and ADMM algorithms can find exact results and are more robust, and ADMM gives a lower transport cost in general.

*Notations.* We use the notations $\alpha_d = (\alpha, \cdots, \alpha) \in \mathbb{R}^d$ and $\alpha_{d \times d}$ for the constant $d \times d$ matrix. The transport cost of a map $T$, which pushes distribution $\mu$ to $\nu$, is defined to be $\mathbb{E}_{x \sim \mu}[|x - Tx|^2]$.

## 2 THE MONGE PROBLEM AS CONSTRAINT OPTIMIZATION

### 2.1 THE MONGE PROBLEM

Let $(X, \mu)$, $(Y, \nu)$ be two separable metric probability spaces. The *Monge problem* is to find a transport map $T : X \mapsto Y$ that realizes the infimum

$$\inf \left\{ \int_X c(x, Tx) d\mu(x) \middle| T_{\#}\mu = \nu \right\} \tag{1}$$

where $T_{\#}\mu$ denotes the push forward of $\mu$ and $c : X \times Y \to \mathbb{R}_+$ is a Borel measurable function which is lower semicontinuous. In this paper, we simply take the distance $|x - y|^2$ but our method applies to other distance functions.

The existence of the Monge problem is difficult and does not hold always. However, under suitable conditions, for example for continuous distributions without atoms, the existence and uniqueness of the Monge problem is guaranteed (see for example, (Villani, 2009, Theorem 5.30). Therefore, here we focus on learning transport maps between continuous distributions. For discrete distributions, one can apply dequantization techniques to transform them to continuous distributions Ho et al. (2019).

### 2.2 THE MONGE PROBLEM AS CONSTRAINT OPTIMIZATION

In order to solve the Monge problem, we use a generative network, denoted by $T_\theta$ with parameter set $\theta$, which inputs random samples from the distribution $\mu$ and generates samples representing the target distributions $\nu$. As can be seen from the definition, the Monge problem is a constraint optimization problem. However, the constraint $T_{\#}\mu = \nu$ is a highly nonlinear constraint. In order to impose this constraint, we take $\mathfrak{d}(\cdot|\cdot)$ to be a distance function (such as the Wasserstein distance, the MMD (Gretton et al. (2012)) or the IPM (Müller (1997))) or a probability divergence (such as the Kullback–Leibler (KL) divergence). The constraint optimization problem reads as

$$\min_\theta \mathbb{E}_{x \sim \mu} \left[ |x - T_\theta x|^2 \right], \quad \text{s.t.} \quad \mathfrak{d}(T_{\theta \#}\mu|\nu) = 0. \tag{2}$$

The objective of this paper is to solve the above problem using techniques from the constraint optimization theory (Bertsekas (2014)).

Since a neural network may not fully reveal the target distributions $\nu$, the above problem can be relaxed to

$$\min_\theta \mathbb{E}_{x \sim \mu} \left[ |x - T_\theta x|^2 \right], \quad \text{s.t.} \quad \mathfrak{d}(T_{\theta\#}\mu|\nu) \leq \alpha, \tag{3}$$

When $\alpha$ goes towards zero, we can prove that the solution $T_{\theta_\alpha}$ to the problem (3) converges to the solution of the original Monge problem (1). The following theorem holds:

**Theorem 1.** *Let $\mu, \nu$ be two probability measures on $\mathbb{R}^d$ with finite second moments and are absolutely continuous. Let $T_\theta$ be given by a neural network with bounded width(each with at least $2d + 2$ neurons) and arbitrary depth, and with non-affine activation functions. Suppose for any $\alpha > 0$, there exists a solution $\theta_\alpha^*$ to problem (2), then as $\alpha \to 0$, $T_{\theta_\alpha^*} \to T$ where $T$ is a solution of the Monge problem (1). Moreover, $\sup_{x \in \mathcal{X}} |T_\alpha(x) - T(x)|_{\mathcal{C}} \leq C\tilde{\alpha}$ for some constant $C$ for any compact subset $\mathcal{X} \in \mathbb{R}^d$.*

Proof of the above theorem follows from the universal approximation theorem (Kidger & Lyons (2020)) and the existence theorem of the Monge problem (Villani (2009)), and is given in Appendix A.1.

### 2.3 EXAMPLE: THE MONGE PROBLEM FROM GAUSSIAN TO GAUSSIAN

For the case when $\mu \sim \mathcal{N}(X_1, \Sigma_1)$ and $\nu \in \mathcal{N}(X_2, \Sigma_2)$ are two multivariate normal distributions, the optimal transport map is unique and can be explicitly given by $T^* : x \mapsto X_2 + A^*(x - X_1)$ with $A^* = \Sigma_1^{-1/2}(\Sigma_1^{1/2}\Sigma_2\Sigma_1^{1/2})^{1/2}\Sigma_1^{-1/2}$ (Olkin & Pukelsheim (1982)). Taking $\mathfrak{d} = D_{\text{KL}}$ to be the KL-divergence, we prove that the solution to problem (2) with $T_\theta x = Ax + b$ ($\theta = \{A \in \mathbb{R}^{d \times d}, b \in \mathbb{R}^d\}$) is the optimal transport map $T^*$ (Theorem 2 in Appendix A.1).

## 3 CONSTRAINT OPTIMIZATION FOR OPTIMAL TRANSPORT

We propose to leverage three different algorithms to solve the constraint problem (2).

### 3.1 PENALTY METHOD (OTCOP-P)

*Standard Lagrangian (SL).* We introduce a Lagrangian multiplier $\lambda$ and take the Lagrangian function as

$$\mathcal{L}_{SL}(\theta, \lambda) = \mathbb{E}_{x \sim \mu} \left[ |x - T_\theta x|^2 \right] + \lambda\mathfrak{d}(T_{\theta\#}\mu|\nu). \tag{4}$$

Then the solution to the problem (2) is a saddle point of the above Lagrangian. By duality theory, for each $\alpha \geq 0$, the problem (3) corresponds to the duality problem $\min_\theta \mathcal{L}_{SL}(\theta, \lambda, 0)$ for a $\lambda \in [0, \infty]$. Hence we can take a suitable $\lambda$ to solve the problem (2) approximately.

According to the Brenier's polar factorization theorem Brenier (1991), the optimal transport map should satisfy $\nabla \times T = 0$, hence we can add an additional term $|\nabla \times T|^2$ into the above Lagrangian to impose this constraint. We will show experimentally that without this term, this constraint is almost satisfied and we will not included in our implementations.

*Quadratic penalty (QP).* Instead of taking $\mathfrak{d}(T_{\theta\#}|\nu)$, we can take a quadratic penalty loss

$$\mathcal{L}_{QP}(\theta, \rho) = \mathbb{E}_{x \sim \mu} \left[ |x - T_\theta x|^2 \right] + \frac{1}{2}\rho(\mathfrak{d}(T_{\theta\#}\mu|\nu))^2. \tag{5}$$

As $\rho$ goes towards infinity, the constraint violations is penalized with increasing severity. For example, we can take $\rho_k$ at the $k$th training step to be increased by multiplying by a constant bigger than 1 and parameter $\theta$ can be updated using gradient descent considering $\rho$ as a constant.

*Convergence.* Suppose there exists a global minimizer to the problem (2), and $\theta_k$ is the exact minimizer of $\mathcal{L}_{QP}(\theta, \rho_k)$ and $\rho_k \uparrow \infty$. Then any limit point of the sequence $\{\theta_k\}$ is a solution to problem (2). Moreover, for any $\varepsilon > 0$, there exists a sufficient large $K > 0$, $|\theta_k - \theta^*| \leq \varepsilon$ for $k \geq K$ (see (Nocedal & Wright, 1999, Theorem 17.1)). In addition, without assuming a global minimizer, for a sequence $\theta_k$ such that $\nabla_{\theta_k}\mathcal{L}_{QP}(\theta_k; \rho_k) \to 0$, its all limit points $\theta^*$ satisfy the Karush–Kuhn–Tucker (KKT) conditions and there exists a subsequence such that $\lim_{k \to \infty}(\rho_k\mathfrak{d}(T_{\theta_k\#}|\nu) = \lambda^*$, where $\lambda^*$

is the multiplier that satisfies the KKT condition (see Appendix A.4 for the KKT condition and see (Nocedal & Wright, 1999, Theorem 17.2) for the proof).

*Advantages and disadvantages.* The penalty method is simple and easy to implement. However, since the optimal value of the Lagrangian multiplier $\lambda$ is unkown (SL) or the optimal condition for the Lagrangian multiplier $\rho$ is infinite (QP), the penalty always introduce errors and the exact solution to the problem (2) cannot be reached. Moreover, the Hessian of the Lagrangian $\nabla^2_{\theta\theta}\mathcal{L}_{QP}$ becomes singular as $\rho$ goes towards infinity and cause ill-condition problems. These issues can be solved by the methods below, but at the expense of a more computational cost.

## 3.2 THE AUGMENTED LAGRANGIAN METHOD (OCTOP-AL)

In order to overcome the above issues, we can use the augmented Lagrangian method, by taking the loss function as

$$\mathcal{L}_{AL}(\theta, \lambda, \rho) = \mathbb{E}_{x\sim\mu}\left[|x - T_\theta x|^2\right] + \lambda\mathfrak{d}(T_{\theta\#}\mu|\nu) + \frac{\rho}{2}(\mathfrak{d}(T_{\theta\#}\mu|\nu))^2. \tag{6}$$

The above function combines standard Lagrangian penalty (4) and quadratic Lagrangian penalty (5). At the $k$th iteration, fix $\lambda_k, \rho_k$ and solve $\theta_k = \arg\min_\theta \mathcal{L}_{AL}(\theta, \lambda_k, \rho_k)$. After the minimization, we update $\lambda$ by $\lambda_{k+1} = \lambda_k + \rho_k\mathfrak{d}(T_{\theta_k\#}\mu|\nu)$. Comparing the KKT conditions of the SL and the AL (see Appendix A.4) implies $\lambda_k + \rho_k\mathfrak{d}(T_{\theta_k\#}\mu|\nu) \approx \lambda^*$ when $\lambda_k$ is taken close to $\lambda^*$. Hence $\mathfrak{d}(T_{\theta_k\#}\mu|\nu) \approx (\lambda^* - \lambda^k)/\rho$. Compared to the quadratic penalty method that $\mathfrak{d}(T_{\theta_k\#}\mu|\nu) \approx \lambda^*/\rho$, the infeasibility in $\theta_k$ will be much smaller. Moreover, for certain choice of $\rho$, the local solution of (2) is a strict local minimizer of $\mathcal{L}_{AL}(\theta, \lambda, \rho)$ ((Nocedal & Wright, 1999, Chapter 17)).

*Convergence.* One of the nice properties of the AL method is that for the exact Lagrangian multiplier $\lambda^*$, the solution $\theta^*$ of the problem (2) is a strict minimizer of $\mathcal{L}_{AL}(\theta, \lambda^*, \rho)$ for all $\rho$ sufficiently large. The existence of a threshold is proved under the condition that $\nabla^2_\theta\mathcal{L}_{SL}(\theta^*, \lambda^*)$ is locally strictly positive ((Nocedal & Wright, 1999, Theorem 17.6)). Thus we can take $\rho$ to be increasing at each minimizing step and when $\rho$ becomes bigger than some threshold value $\bar{\rho}$, gradient descent methods could find the local minimizer around $\theta^*$.

*Advantages and disadvantages.* AL method introduces the multiplier estimates and reduces the likelihood that large values of $\rho$ will be needed to obtain good feasibility and accuracy. The method is also simple and easy to implement. However, since this is a min-max method, training may experience oscillations and slower convergence rates.

## 3.3 ADMM METHOD (OCTOP-ADMM)

The ADMM method blends the decomposition techniques and the AL method and provides an efficient way for constraint optimizations Boyd et al. (2011). Let $S$ be the set $S = \{T_\theta : T_{\theta\#}\mu = \nu\}$, problem (2) can be rewritten into the form

$$\min_\theta \mathbb{E}_{x\sim\mu}\left[|x - T_\theta x|^2\right] + \mathbf{1}_S(T_\theta), \tag{7}$$

where $\mathbf{1}_S$ is the indicator function that equals 0 if $T_\theta \in S$ and equals $\infty$ if $T_\theta \notin S$. In order to apply the ADMM method, we rewrite the above problem into the form

$$\min_{\theta_1, \theta_2} \mathbb{E}_{x\sim\mu}\left[|x - T_{\theta_1}x|^2\right] + \mathbf{1}_S(T_{\theta_2}), \quad \text{s.t.} \quad T_{\theta_1} = T_{\theta_2}. \tag{8}$$

In the ADMM method, we alternatively update $\theta_1$ and $\theta_2$. First we take $\theta_2$ to be constant and take the minimization of the above problem over $\theta_1$, and then we project $\theta_2$ onto the space $S$. In detail, we introduce the loss function

$$\begin{aligned}\mathcal{L}_{ADMM}(\theta_1, \theta_2, \Lambda, \rho) = \ &\mathbb{E}_{x\sim\mu}\left[|x - T_{\theta_1}x|^2\right] + \mathbf{1}_{\mathfrak{d}(T_{\theta_2\#}\mu|\nu)=0} \\ &+ \Lambda^T(T_{\theta_1}x - T_{\theta_2}x) + \frac{\rho}{2}(\mathbb{E}_{x\sim\mu}[|T_{\theta_1}x - T_{\theta_2}x|^2]).\end{aligned} \tag{9}$$

Here $\Lambda \in \mathbb{R}^d$ is the multiplier. The training procedure is given by

1. $\theta_1^{k+1} = \arg\min_{\theta_1} \mathcal{L}_{ADMM}(\theta_1, \theta_2^k, \Lambda^k, \rho)$ (assuming $\mathbf{1}_{\mathfrak{d}(T_{\theta_2\#}\mu|\nu)=0} = 0$);

2. $\theta_2^{k+1} = \arg\min_{\theta_2} \mathfrak{d}(T_{\theta_2\#}\mu|\nu)$;

3. $\Lambda_{k+1} = \Lambda_k + \rho\mathbb{E}_{x\sim\mu}(T_{\theta_1^{k+1}\#}(x) - T_{\theta_2^{k+1}\#}(x))$.

*Convergence.* The convergence of ADMM method is only known to hold under convex conditions or for some non-convex problems (Boyd et al. (2011)). Using results of Wang et al. (2019), we can prove the convergence of the ADMM method if we modify the above method by relaxing problem (8) to

$$\min_\theta \mathbb{E}_{x\sim\mu}\left[|x - T_{\theta_1}x|^2\right] + \eta_\varepsilon(\mathfrak{d}(T_{\theta_2}, S)), \quad \text{s.t.} \quad \theta_1 = \theta_2, \tag{10}$$

where $\eta_\varepsilon$ is the mollifier function which converges to the $\delta$-function as $\varepsilon \to 0$. We show that the ADMM method converges to the KKT points and if the corresponding Lagrangian is a Kurdyka-Łojasiewicz function, the ADMM method converges globally to the unique solution (see Appendix A.3 for details). As a consequence, there exists a convergence sequence of the ADMM method for the problem (8) approximately.

*Advantage and disadvantages.* The advantage of ADMM is that it decomposes the Monge problem into two sub-problems: minimizing the transport cost and minimizing the $D_{\mathrm{KL}}$. Compared to the AL method, AMDD solves two decomposed minimization problem and AMDD may also converge faster than the AL method in some situations (Wang et al. (2019)). However, the method also solves a min-max problem and may facing oscillations in the training process.

## 3.4 THE DISTANCE $\mathfrak{d}$

We need a distance/divergence functional $\mathfrak{d}$ to compare the generated distribution and the target distribution. Here we require $\mathfrak{d}$ to be able to compute using samples.

*The KL divergence $D_{\mathrm{KL}}$.* When the densities of the target distribution is known, we can use networks of normalizing flow (Rezende & Mohamed (2015)) as $T_\theta$ and the $D_{\mathrm{KL}}$ is computed via

$$D_{\mathrm{KL}}(T_{\theta\#}\mu|\nu) = \mathbb{E}_{x\sim\mu}[\log\mu(x) - \log\det J_{T_\theta}(x) - \log\nu(T_\theta x)], \tag{11}$$

where $J_{T_\theta}$ is the Jacobian of the transport map which is able to compute using the normalizing flow networks. When the target densities are unknown. We need to reverse the KL divergence and take

$$D_{\mathrm{KL}}(\nu|T_{\theta\#}\mu) = \mathbb{E}_{x\sim\nu}[\log\nu(x) - \log\mu(T_\theta^{-1}x) - \log\det J_{T_\theta}(T^{-1}x)].$$

We can drop the first term in the bracket since it is a constant. We can take $\mathcal{L}_{SL}$ in (4) as

$$\mathcal{L}_{SL}(\theta, \lambda) = \mathbb{E}_{x\sim\mu}\left[|x - T_\theta x|^2\right] + \lambda\mathbb{E}_{x\sim\nu}[-\log\mu(T_\theta^{-1}x) - \log\det J_{T_\theta}(T^{-1}x)].$$

For the ADMM method, the second step changes to $\theta_2^k \leftarrow \arg\min_\theta \mathbb{E}_{x\sim\nu}[-\log\mu(T_\theta^{-1}x) - \log\det J_{T_\theta}(T^{-1}x)]$. In order to apply the AL method, one can use $\mathbb{E}_{x\sim\nu}[-\nabla_\theta\log\mu(T_\theta^{-1}x) - \nabla_\theta\log\det J_{T_\theta}(T^{-1}x)]$ to replace the $D_{\mathrm{KL}}$ terms in (6).

*Test function as multiplier.* A weak form of the constraint $T_{\theta\#}\mu = \nu$ is that for any measurable function $f$,

$$\int f(T_\theta x)d\mu(x) = \int f(x)d\nu(x)$$

we can introduce a discriminator network $f_w$ as Lagrangian multiplier and the Lagrangian becomes

$$\mathcal{L}_{SL}(\theta, w) = \mathbb{E}_{x\sim\mu}[|x - T_\theta x|^2] + \mathbb{E}_{x\sim\mu}[f_w(T_\theta x)] - \mathbb{E}_{z\sim\nu}[f_w(z)]. \tag{12}$$

The augmented Lagrangian then become

$$\begin{aligned}\mathcal{L}_{AL}(\theta, w, \rho) = {} & \mathbb{E}_{x\sim\mu}[|x - T_\theta x|^2] + \mathbb{E}_{x\sim\mu}[f_w(T_\theta x)] - \mathbb{E}_{z\sim\nu}[f_w(z)] \\ & + \frac{\rho}{2}(\mathbb{E}_{x\sim\mu}[f_w(T_\theta x)] - \mathbb{E}_{z\sim\nu}[f_w(z)])^2.\end{aligned} \tag{13}$$

The training procedure is given by

1. $\theta^k = \arg\min_\theta \mathcal{L}_{AL}(\theta, w, \rho)$;

2. $w^{k+1} = w^k + \rho^k\mathbb{E}_{x\sim\mu}[\nabla_w f_w(T_\theta x)] - \mathbb{E}_{z\sim\nu}[\nabla_w f_w(z)]$;

3. Assign $\rho^{k+1} \geq \rho^k$.

To apply the ADMM method, we take the loss function to be

$$\mathcal{L}_{ADMM}(\theta_1, \theta_2, w_1, w_2, \rho) = \mathbb{E}_{x \sim \mu}[|x - T_{\theta_1}x|^2] + \mathfrak{d}_{w_2}(T_{\theta_2\#}\mu|\nu))$$
$$+ \mathbb{E}_{x \sim \mu}[f_{w_1}(T_{\theta_1}x) - f_{w_1}(T_{\theta_2}x)] + \frac{\rho}{2}\mathbb{E}_{x \sim \mu}[f_{w_1}(T_{\theta_1}x) - f_{w_1}(T_{\theta_2}x)]^2. \quad (14)$$

Then the optimization is decoupled into optimization over $\theta_1$ to learn the optimal transport map and the optimization over $\theta_2$ to learn the target distribution. Here we can take for example $\mathfrak{d}_{w_2}$ to be the loss function of Wasserstein GANs with gradient penalty (Gulrajani et al. (2017)):

$$\mathfrak{d}_{w_2}(T_{\theta_2\#}\mu|\nu)) = \mathbb{E}_{x \sim \mu}[f_{w_1}(T_\theta x)] - \mathbb{E}_{z \sim \nu}[\nabla_{w_1} f_{w_2}(z)] + c\mathbb{E}_{\hat{x} \sim \gamma}[(\|\nabla_{\hat{x}} f_{w_2}(\hat{x})\|_2 - 1)^2],$$

where $c$ is a positive constant and $\gamma$ is the linear interpolation of $T_{\theta\#}\mu$ and $\nu$. The training procedure is given by

1. $\theta_1^{k+1} = \arg\min_{\theta_1} \mathcal{L}_{ADMM}(\theta_1, \theta_2^k, w_1^k, w_2^k, \rho^k)$;

2. $(\theta_2^{k+1}, w_2^{k+1}) = \arg\min_{\theta_2} \max_{w_2} \mathcal{L}_{ADMM}(\theta_1^{k+1}, \theta_2, w_1^k, w_2, \rho^k)$ updated same as WGAN with gradient penalty.

3. $w_1^{k+1} = w_1^k + \rho^k \mathbb{E}_{x \sim \mu}[\nabla_{w_1} f_{w_1}(T_\theta x)] - \mathbb{E}_{z \sim \nu}[\nabla_{w_1} f_{w_1}(z)]$.

### 3.5 IMPLEMENTATION OF THE ALGORITHMS

The implementations of the algorithms are given in Algorithm 1, 2, 3. Here we give the implementation when $\mathfrak{d}$ is given directly without using a discriminator network. For the case when test function is used as multiplier, see section 3.4 and Appendix **??** for details.

---

**Algorithm 1** Solving the Monge problem with the penalty method

**Input** Data: $X \sim \mu, Y \sim \nu$, Constants: $\lambda_0$, $\rho_0$, $\alpha > 1$, Training step: $\eta$.
**for** number of training iterations **do**
    **for** $m$ steps **do**
        $\theta \leftarrow \theta - \eta\nabla_\theta \mathcal{L}_{SL}$
        (or $\theta \leftarrow \theta - \eta\nabla_\theta \mathcal{L}_{QP}$)
    **end for**
    Update $\rho \leftarrow \alpha\rho$.
**end for**

---

**Algorithm 2** Solving the Monge problem with the augmented Lagrangian method

**Input** Data: $X \sim \mu, Y \sim \nu$ Constants: $\lambda_0, \rho_0, \alpha > 1$, Training step: $\eta$.
**for** number of training iterations **do**
    **for** $m$ steps **do**
        $\theta \leftarrow \theta - \eta\nabla_\theta \mathcal{L}_{AL}$
    **end for**
    Update $\lambda \leftarrow \lambda + \rho\mathfrak{d}(T_{\theta\#}\mu|\nu)$
    Update $\rho \leftarrow \alpha\rho$.
**end for**

---

**Algorithm 3** Solving the Monge problem with the ADMM

**Input** Data: $X \sim \mu, Y \sim \nu$, Constants: $\lambda_0, \rho_0, \alpha > 1$, Training step: $\eta$.
**for** number of training iterations **do**
    **for** $m_1$ steps **do**
        $\theta_1 \leftarrow \theta_1 - \eta\nabla_\theta \mathcal{L}_{ADMM}(\theta_1, \theta_2, \lambda, \rho)$
    **end for**
    **for** $m_2$ steps **do**
        $\theta_2 \leftarrow \theta_2 - \eta\nabla_\theta \mathfrak{d}(T_{\theta_2\#}\mu|\nu)$
        $w_2 \leftarrow w_2 + \eta\nabla_w \mathfrak{d}(T_{\theta_2\#}\mu|\nu)$ (if $\mathfrak{d} = \mathfrak{d}_w$)
    **end for**
    Update $\lambda \leftarrow \lambda + \rho\mathfrak{d}(T_{\theta_1\#}\mu|T_{\theta_2\#}\mu)$
**end for**

---

## 4 EXPERIMENT

### 4.1 MULTIVARIATE NORMAL DISTRIBUTIONS

*Linear maps.* First, we consider the optimal transport between multivariate normal distributions $\mu = \mathcal{N}(X_1, \Sigma_1)$ and $\nu = \mathcal{N}(X_2, \Sigma_2)$. Considering the map given by $T_\theta = Ax + b$ with $\theta = \{A \in \mathbb{R}^{d \times d}, b \in \mathbb{R}^d\}$, we prove in Theorem 2 that the solution to problem (2) gives the correct solution to the Monge problem. Indeed, in this case, the SL, AL and ADMM algorithms reduce to optimization of linear objective function constraint by a nonlinear function and it could be theoretically analyzed by the constraint optimization theory Bertsekas (2014). Here we take $X_1 = 0_2, \Sigma_1 = I_2$ and $X_2 = 0_2, \Sigma_2 = [[4, 1], [1, 4]]$, let $T_\theta x = Ax$ with $A = [[a, b], [b, a]]$ ($\theta = \{a, b\}$). Then the problem (2) reduces to

$$\min_{a,b} 2(1 - a)^2 + 2b^2,$$

$$\text{s.t. } D_{\mathrm{KL}}(T_\theta \mu | \nu) = \tfrac{1}{30}(-15 \log\left(\frac{1}{15}\left(a^2 - b^2\right)^2\right) + 8a^2 - 4ab + 8b^2 - 30) = 0.$$

The landscapes of $D_{\mathrm{KL}}$ and $\mathcal{L}_{SL}(\theta, \lambda = 1)$ as well as the value of $\min_\theta \mathcal{L}_{SL}(\theta, \lambda)$ as functions of $\lambda$, are plotted in Figure 1. As can be seen from the figure, the function $D_{\mathrm{KL}}(T_{\theta\#}\mu | \nu)$ has multiple minimizers (red points), whereas the Lagrangian $\mathcal{L}_{SP}(\theta, 1)$ has a unique global minimizer (blue point). Hence minimizing the Lagrangian $\mathcal{L}_{SL}$ helps to find the optimal transport maps by finding the approximate map with the lowest transport cost among all maps that realize the target distributions. More importantly, from the figure, the whole domain can be divided into four pieces by the landscape of $D_{\mathrm{KL}}$, and starting in each piece, gradient descent method will converge to one of the four different points. In contrast, the landscape of the Lagrangian changes dramatically and starting from any point, gradient descent method will only converge to one point.

The Langrangian introduces errors for $\lambda$ finite. As can be seen from the leftmost figure, the constraint $D_{\mathrm{KL}} = 0$ is more relaxed as $\lambda$ becomes smaller. However, for large $\lambda$, the barrier region becomes wider and the gradient descent may converge to a local minimizer away from the optimal result. For small $\lambda$, the gradient descent is easy to find the global minimizer, but since penalty introduces error, the relaxation effect of the constraints can also push the minimizer away from the solution of the Monge problem. The choice of $\lambda$ needs to take this tradeoff into considerations. Training using a one layer linear neural network confirms the above analysis.

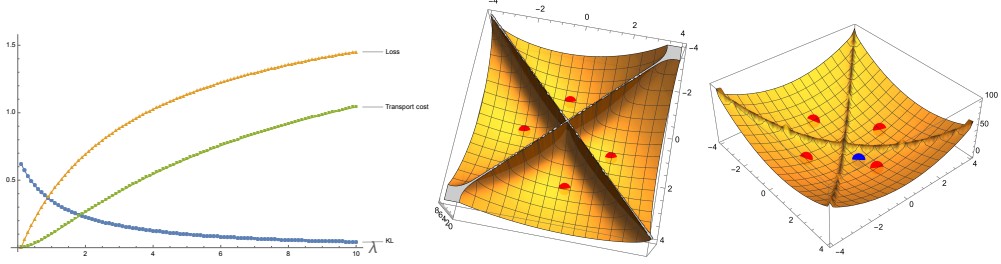

Figure 1: Graph of the KL error and the loss function of the SP method (left: $\min_\theta \mathcal{L}_{AL}(\theta, \lambda)$ as well as the corresponding $D_{\mathrm{KL}}$ and transport cost as functions of $\lambda$; middle: landscape of $D_{\mathrm{KL}}$ as functions of $a, b$; right: landscape of $\mathcal{L}_{SL}(\theta, 1)$ as functions of $a, b$). Red points: minimizers of the $D_{\mathrm{KL}}$, blue point: minimizer of $\mathcal{L}_{SL}(\theta, 1)$.

*Training with neural networks with nonlinear activation function.* Next we present our results on the training of Gaussian to Gaussian distributions with neural networks with nonlinear activation function. We take $X_1 = 0_d, X_1 = 1_d$ and $\Sigma_1 = I_d, \Sigma_2 = 3I_d + 1_{d \times d}$. Theoretical analysis gives that the optimal transport distance is $2d$ (see Theorem 2). Since here we use the $D_{\mathrm{KL}}$ as distance function which is always positive, so the QP method behaves similarly as the SL method with different multiplier. Hence the results of the QP method is not presented here. The results are given in Table 1. and the 784D Gaussian is taken for $X_1 = 0_{784}, X_1 = 2 \cdot 1_{784}$ with $\Sigma_1 = \Sigma_2 = I_{784}$ (theoretical result of the optimal transport distance is $784 * 4$). As can be seen from the table, all three algorithms give nice result and learn approximately the optimal transport map with a high

Table 1: Training results on Gaussian and Gaussian mixtures

| benchmark | method | $D_{\mathrm{KL}}$ | Transport cost/$d$ | benchmark | $D_{\mathrm{KL}}$ | Transport cost/$d$ |
|---|---|---|---|---|---|---|
| 2D Gaussian | SL | 0.002 | 2.018 | 78D Gaussian | 0.228 | 1.866 |
| | AL | 0.002 | 2.021 | | 0.260 | 2.290 |
| | ADMM | 0.003 | 1.950 | | 0.805 | 2.250 |
| 784D Gaussian | SL | 0.365 | 3.981 | 2D mixture | 0.034 | 0.048 |
| | AL | 0.333 | 4.001 | | 0.021 | 0.066 |
| | ADMM | 0.399 | 3.998 | | 0.059 | 0.035 |

accuracy. Training is done by using a $d$ width and 10 depth neural network with $\mathrm{tanh}$ activation for all cases except for 78D Gaussian, for which a 100 depth neural network is used in order to learn the correlations correctly. Remarkably, for the highly correlated distributions in high dimensions, our method gives a nice result.

*ADMM learns a lower transport cost.* From the figure, we can see that the ADMM method learns a lower transport cost compared to other methods. This is because during training, minimizing the $D_{\mathrm{KL}}$ between the generated and target distribution may not converge to the solution to the Monge problem, as illustrated in the simple 2D case above. Hence, the splitting feature enables ADMM to learn a transport map with lower transport cost. This can also be seen from the learning curve in Figure 4 in Appendix A.5. The transport of the second network ($T_{\theta_2}$) has a higher transport cost when learning the target distribution, whereas the first network ($T_{\theta_1}$ has a lower transport cost, while the learned target distribution remains accurate.

## 4.2 GAUSSIAN TO GAUSSIAN MIXTURES

We take a four component Gaussian mixture, each with variance matrix $0.5I_2$, centers lying on the four corners of the square $[-1,1]^2$ and learn the optimal transport map between two dimensional standard Gaussian to this mixture distribution. We plot the Jacobian graph of the learned map and the value of the $D_{\mathrm{KL}}$ between target distribution and the prediction by the network in Figure 2. As can be seen from the figure, solely minimizing the $D_{\mathrm{KL}}$ does not fulfil the right directions of the optimal transport maps. Using our method, the learned map is more balanced and approximately satisfies the condition $\nabla \times Tx = 0$. Note that here we donot include the penalty of $|\nabla \times Tx|^2$. From the learning curves, we confirm the findings above that ADMM learns a lower transport cost than the other methods. Here the transport cost of the second network converges to around 0.07, while the transport cost of the first network converges to about one half of that of the second network.

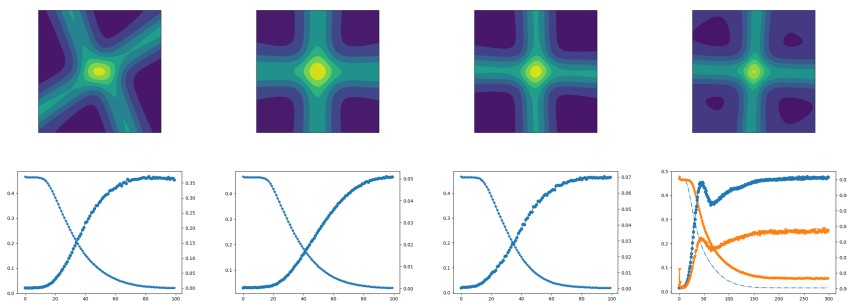

Figure 2: Graph of the KL error and the loss function of the minimization of $D_{\mathrm{KL}}$, $\min D_{\mathrm{KL}}$, the SP, AL and the ADMM method (from left to right). The $D_{\mathrm{KL}}$ curve is the decreasing line and the transport cost is the increasing line. For the bottom right figure, orange line is for the first network and blue line is for the second network

*AL and ADMM methods are more robust.* Compared to the SL, AL gives a more robust result with respect to the value of $\lambda$. For a well chosen $\lambda$, SL performs as good as AL. However, if $\lambda$ is not taken properly, the obtained transport cost will be higher or the target distribution is not well

realized. For ADMM method, we can see the choice of $\rho$ affects the training process, but in a big range, the choice of $\rho$ has little effects on the final result (see appendix A.5 for the graphs indicating this finding). This confirms the benefits of the AL and ADMM method in the literature (Nocedal & Wright (1999); Boyd et al. (2011)).

### 4.3 IMPROVED TRANSPORT COST OF GANS

We use the test function as multiplier and test the performance of the algorithms described in section 3.4.

*The SL method help overcome the vanishing gradient issue.* The constraint $f_w$ is a Lipschitz-1 function is important in WGAN. Without this constraint, the training of WGAN may face vanishing gradients issue, as illustrated in Figure 6 in Appendix A.6. However, adding the transport cost to the cost function as in (12), the training no longer face this difficulty and the target distribution could be learned. Note that here we donot use a penalty term on the discriminator network as the gradient penalty method, the transport cost is only a function of the generator network and no regularization of the discriminator network is needed.

*All three methods significantly reduce the transport cost of GAN.* Compared to the WGAN-GP, which has a transport cost around 3.68, all three methods (SL, AL, ADMM) described in section 3.4 show a significantly lower transport cost (0.077 for SL, 0.071 for AL and 0.049 for the ADMM). The transport losses are plotted in Figure 7 in Appendix A.6. However, the SL method is sensitive to the choice of $\lambda$, while the AL and ADMM methods are more robust.

*MNIST.* We also train the WGANs on the MNIST dataset. By using WGAN with SL($\lambda = 1$), we obtain a transport cost around 1.81 compared to 1.87 with only WGAN. MNIST like samples generated by the learned optimal is plotted in Figure 3. Therefore, our method lowers the transport cost of WGAN while keeping the qualify of the generated distributions.

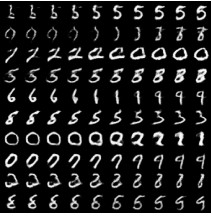

Figure 3: MNIST like samples generated by WGAN with SL penalty

## 5 CONCLUSION

We have shown that the incorporation of constraint optimization tools provides a direct and efficient way for computing optimal transport maps. By solving the Monge problem directly, our method avoids using special network structures or solving the dual problem. Moreover, applying our method to WGAN shows a lower transport cost for the generative networks without sacrificing the quality of the generated data.

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

## A APPENDIX

### A.1 PROOF OF THEOREM 1

The proof of Theorem 1 is given below by combing the existence theorem for the Monge problem and the universal approximation of neural networks.

*Proof.* First given two probability measures $\mu, \nu$ satisfying the assumptions of Theorem 1, then (Villani, 2009, Theorem 5.20) implies that there exists a unique deterministic solution $T$ to the Monge problem and the map $T$ is a Borel map. Then by Lusin's theorem, there exists a continuous function $\tilde{T} \in C(\mathbb{R}^d)$ such that $\tilde{T} = T$ on any compact subset $\mathcal{X} \subset \mathbb{R}^d$ almost everywhere. Since $T_\theta$ is given by a arbitrary depth neural network, we can use the universal approximation theorem in Kidger & Lyons (2020) to conclude:

*For any $\varepsilon > 0$, there exists $T_{\hat{\theta}}$ such that* $\sup_{x \in \mathcal{X}} |T_{\hat{\theta}}(x) - \tilde{T}| \leq \varepsilon$.

Therefore, $\mathfrak{d}(T_{\hat{\theta}\#}\mu|\nu) = \mathfrak{d}(T_{\hat{\theta}\#}\mu|T_{\#}\mu) \leq \mathfrak{d}(T_{\hat{\theta}\#}\mu|\tilde{T}_{\#}\mu) + \mathfrak{d}(T_{\#}\mu|\tilde{T}_{\#}\mu) \leq 2\varepsilon$.

Fix $\alpha$ and take $\varepsilon < \alpha/2$, since $\mathfrak{d}(T_{\hat{\theta}\#}\mu|\nu) \leq \alpha$, we can get that the solution $T_{\theta_\alpha^*}$ to the problem (3) satisfies

$$\mathbb{E}_{x\sim\mu}[|x - T_{\theta_\alpha^*}x|^2] \leq \mathbb{E}_{x\sim\mu}[|x - T_{\hat{\theta}}x|^2] \leq \mathbb{E}_{x\sim\mu}[|x - Tx|^2] + \varepsilon. \tag{15}$$

Let $\{\alpha_k\}$ be a sequence converging to zero and $T_{\theta_k}$ be the solution to the corresponding problem (3), then $\mathfrak{d}(T_{\theta_k\#}\mu|\nu) \leq \alpha_k \to 0$ implies $T_{\theta_k\#}\mu \to T_{\theta^*\#}\mu = T_{\#}\mu$. Since $T$ solves the problem (1), we have

$$\mathbb{E}_{x\sim\mu}[|x - T_\theta x|^2] \leq \mathbb{E}_{x\sim\mu}[|x - T_{\theta^*}x|^2].$$

On the other hand, by taking limit $\alpha \to 0$ in (15), we have $\mathbb{E}_{x\sim\mu}[|x - T_{\theta^*}x|^2] \leq \mathbb{E}_{x\sim\mu}[||x - T_\theta x|^2]$. Combing this with the previous inequality implies that $\mathbb{E}_{x\sim\mu}[|x - T_\theta x|^2] = \mathbb{E}_{x\sim\mu}[||x - T_{\theta^*}x|^2]$. By the unique existence of the Monge map, $T_{\theta^*} = T$. Moreover, we have by the convergence of $T_{\theta_\alpha} \to T_{\theta^*}$

$$\mathbb{E}_{x\sim\mu}[|T_{\theta_\alpha}x - Tx|^2] = \mathbb{E}_{x\sim\mu}[|T_{\theta_\alpha}x - T_{\theta^*}x|^2] \leq C\alpha. \tag{16}$$

$\square$

A.2 PROOF OF THEOREM 2

**Theorem 2.** *Let $\mu = \mathcal{N}(X_0, \Sigma_0)$ and $\nu = \mathcal{N}(X_1, \Sigma_1)$ be two multivariate normal distributions with $X_0, X_1 \in \mathbb{R}^d$ and $\Sigma_0, \Sigma_1 \in \mathbb{R}^{d \times d}$. Let $T_\theta : \mathbb{R}^d \mapsto \mathbb{R}^d$ be an linear operator defined by $y = Ax + b$ with $\theta = (A, b)$ and $A \in \mathbb{R}^{d \times d}$, $b \in \mathbb{R}^d$. Then there exists a unique solution to the following problem*

$$\theta^* = \arg\min_\theta \left\{ \int_{\mathbb{R}^d} |x - T_\theta x|^2 d\mu(x), \quad s.t. \quad D_{\mathrm{KL}}(T_{\theta \# \mu} | \nu) = 0 \right\}, \tag{17}$$

*and the corresponding transport map $T_{\theta^*}$ is the optimal transport map in the sense of (1).*

*Proof.* For $x \sim \mu$, the linear transformation $y = Ax + b$ also satisfies a multivariate normal distribution, given by

$$y \sim \rho = \mathcal{N}(AX_0 + b, A\Sigma_0 A^T).$$

Using the formula for KL divergence between two multivariate normal distributions, we can get that the KL-divergence between $\rho$ and $\nu$ is

$$D_{\mathrm{KL}}(\rho|\nu) = \frac{1}{2} \left[ \mathrm{tr}(\Sigma_1^{-1} A\Sigma_0 A^T) + (X_1 - (AX_0 + b))^T \Sigma_1^{-1}(X_1 - (AX_0 + b)) - d \right.$$
$$\left. + \log \frac{\det(A\Sigma_0 A^T)}{\det \Sigma_1} \right] \tag{18}$$

Due to the fact that $\mathrm{tr}B - \log \det B + d \geq 0$ for any positive definite matrix $B > 0$ and $(X_1 - (AX_0 + b))^T \Sigma_1^{-1}(X_1 - (AX_0 + b)) \geq 0$, $\mathrm{KL}(\rho|\nu) = 0$ is equivalent to

$$\Sigma_1^{-1} A\Sigma_0 A^T = I_d, \quad (X_1 - (AX_0 + b)) = 0. \tag{19}$$

The objective function in (17) can be calculated by

$$\int_{\mathbb{R}^d} |x - T_\theta x|^2 d\mu(x) = \int_{\mathbb{R}^d} |x - (Ax + b)|^2 d\mu(x)$$
$$= \mathbb{E}_{x \sim \mu} \left[ ((I - A)x - b)^T ((I - A)x - b) \right]$$
$$= \mathrm{tr}((I - A)^T (I - A) \mathbb{E}_{x \sim \mu_0}(xx^T)) - 2b^T(I - A)X_0 + b^T b$$
$$= \mathrm{tr}((I - A)^T (I - A)(\Sigma_0 + X_0 X_0^T)) - 2b^T(I - A)X_0 + b^T b$$
$$= \mathrm{tr}((I - A)^T (I - A)\Sigma_0) + |(I - A)X_0 - b|^2. \tag{20}$$

Therefore, the optimization problem (17) becomes

$$\min_{A \in \mathbb{R}^{d \times d}, b \in \mathbb{R}^d} \mathrm{tr}((I - A)^T(I - A)\Sigma_0) + |(I - A)X_0 - b|^2,$$
$$\text{s.t.} \quad \Sigma_1^{-1} A\Sigma_0 A^T = I_d, \quad (X_1 - (AX_0 + b)) = 0. \tag{21}$$

Eliminating $b$, this is equivalent to

$$\min_{A \in \mathbb{R}^{d \times d}} \mathrm{tr}((I - A)^T(I - A)\Sigma_0) + |X_0 - X_1|^2,$$
$$\text{s.t.} \quad \Sigma_1^{-1} A\Sigma_0 A^T = I_d. \tag{22}$$

Since $\mathrm{tr}((I - A)^T(I - A)\Sigma_0) = \mathrm{tr}(\Sigma_0) - 2\mathrm{tr}(A^T \Sigma_0) + \mathrm{tr}(A\Sigma_0 A^T) = \mathrm{tr}(\Sigma_0) - 2\mathrm{tr}(A^T \Sigma_0) + d$, the above problem is equivalent to

$$\max_{A \in \mathbb{R}^{d \times d}} \mathrm{tr}(A^T \Sigma_0),$$
$$\text{s.t.} \quad A\Sigma_0 A^T = \Sigma_1. \tag{23}$$

Let $R = \Sigma_0^{\frac{1}{2}} A^T \Sigma_1^{-\frac{1}{2}}$, the above problem could be rewritten as

$$\max_{R \in \mathbb{R}^{d \times d}; R^T R = I_d} \mathrm{tr}(R\Sigma_1^{\frac{1}{2}} \Sigma_0^{\frac{1}{2}}). \tag{24}$$

Note that the optimization is over all orthogonal matrix in $\mathbb{R}^{d \times d}$. The solution is given by $R^* = V^T$ where $\Sigma_1^{\frac{1}{2}} \Sigma_0^{\frac{1}{2}} = VP$ is the polar decomposition of $\Sigma_1^{\frac{1}{2}} \Sigma_0^{\frac{1}{2}}$ and is given by $V = \Sigma_1^{\frac{1}{2}} \Sigma_0^{\frac{1}{2}} ((\Sigma_1^{\frac{1}{2}} \Sigma_0^{\frac{1}{2}})^T \Sigma_1^{\frac{1}{2}} \Sigma_0^{\frac{1}{2}})^{-\frac{1}{2}} = \Sigma_1^{\frac{1}{2}} \Sigma_0^{\frac{1}{2}} (\Sigma_0^{\frac{1}{2}} \Sigma_1 \Sigma_0^{\frac{1}{2}})^{-\frac{1}{2}}$ and $P = (\Sigma_0^{\frac{1}{2}} \Sigma_1 \Sigma_0^{\frac{1}{2}})^{\frac{1}{2}}$ Recalling $R^* = \Sigma_0^{\frac{1}{2}} (A^*)^T \Sigma_1^{-\frac{1}{2}}$, we obtain

$$A^* = \Sigma_0^{-\frac{1}{2}} (\Sigma_0^{\frac{1}{2}} \Sigma_1 \Sigma_0^{\frac{1}{2}})^{\frac{1}{2}} \Sigma_0^{-\frac{1}{2}}, \tag{25}$$

and by the constraint $(X_1 - AX_0 - b) = 0$, we get

$$b^* = A^* X_0 - X_1. \tag{26}$$

The above formula $(A^*, b^*)$ is the same as that obtained by analytical method for the optimal transport and is shown to be the unique solution to the Monge problem (1). $\qquad \square$

### A.3 Proof of the convergence of the ADMM method

**Theorem 3.** *Assume $T_\theta$ is given by a neural network with Lipschitz continuous activation functions and satisfying $\|T_\theta\| \to \infty$ for $\theta \to \infty$. For sufficiently large $\rho$, the ADMM method generates a bounded sequence that converges to the stationary point of the Lagrangian*

$$\mathcal{L}_\varepsilon(\theta_1, \theta_2, \Lambda, \rho) = \mathbb{E}_{x \sim \mu} \left[ |x - T_{\theta_1} x|^2 \right] + \eta_\varepsilon(\mathfrak{d}(T_{\theta_2 \#} \mu | \nu))$$
$$+ \Lambda^T (T_{\theta_1} x - T_{\theta_2} x) + \frac{\rho}{2} (\mathbb{E}_{x \sim \mu}[\Lambda^T (T_{\theta_1} x - T_{\theta_2} x)])^2. \tag{27}$$

*Proof.* We only need to check the conditions in the reference Wang et al. (2019). First we show the objective function is coercive. For bounded $\theta_1 = \theta_2$ and $\theta_1 \to \infty$, then $\mathbb{E}_{x \sim \mu} \left[ |x - T_{\theta_1} x|^2 \right] + \eta_\varepsilon(\mathfrak{d}(T_{\theta_2 \#} \mu | \nu)) = \infty$ and thus is coercive. The Lipschitz continuous assumption implies the objective function is also Lipschitz continuous. Moreover, the solution to the subproblem $\min_{\theta_1} \mathbb{E}_{x \sim \mu} \left[ |x - T_{\theta_1} x|^2 \right]$ and $\min_{\theta_2} \eta_\varepsilon(\mathfrak{d}(T_{\theta_2 \#} \mu | \nu))$ is also Lipschitz continuous. Thus the conditions in Wang et al. (2019) holds and we can conclude that the ADMM method converges. $\qquad \square$

### A.4 The KKT conditions

The KKT condition is the conditions for the saddle point of the Lagrangian. *The KKT condition for the SL Lagrangian (4).* At the saddle point $(\theta^*, \lambda^*)$, the KKT conditions for (4) are

$$\begin{cases} \nabla_\theta \mathcal{L}_{SP}(\theta^*, \lambda^*) = \nabla_\theta \mathbb{E}_{x \sim \mu}([|x - T_\theta x|^2])(\theta^*) + \lambda^*(\nabla_\theta \mathfrak{d}(T_{\theta \#} \mu | \nu))(\theta^*) = 0, \\ \nabla_\lambda \mathcal{L}_{SP}(\theta^*, \lambda^*) = \mathfrak{d}(T_{\theta^* \#} \mu | \nu) = 0. \end{cases}$$

At each training step (taking $\rho$ to be constant), the KKT conditions for the QP loss (5) are

$$\nabla_\theta \mathcal{L}_{QP}(\theta_k, \rho_k) = \nabla_\theta \mathbb{E}_{x \sim \mu}([|x - T_\theta x|^2])(\theta_k) + \rho_k \mathfrak{d}(T_{\theta_k \#} \mu | \nu)(\nabla_\theta \mathfrak{d}(T_{\theta \#} \mu | \nu))(\theta_k) = 0,$$

At each training step (taking $\rho$ to be constant), the KKT conditions for the AL loss (6) are

$$\begin{cases} \nabla_\theta \mathcal{L}_{AL}(\theta_k, \lambda_k, \rho_k) = \nabla_\theta \mathbb{E}_{x \sim \mu}([|x - T_\theta x|^2])(\theta_k) + (\lambda_k + \rho_k \mathfrak{d}(T_{\theta_k \#} \mu | \nu))(\nabla_\theta \mathfrak{d}(T_{\theta \#} \mu | \nu))(\theta_k) = 0, \\ \nabla_\lambda \mathcal{L}_{AL}(\theta_k, \lambda_k, \rho_k) = \mathfrak{d}(T_{\theta^k} \mu | \nu) = 0. \end{cases}$$

### A.5 Training results

The results of 78D Gaussian is plotted below.

We demonstrate the robustness of AL and ADMM method compared to the SL method. The training curves for these three methods are plotted in Figure 5.

### A.6 Experiment details

Code for the numerical experiments is available at https://github.com/otcop/otcop.git.

Table 1 and Figure 2, 3 are produced using a normalizing flow network with planar transformation layers. The initial and target densities are known and samples are drawn from these distributions. Algorithms described in sections 3.1-3.3 are used. The KL divergence is computed via (11).

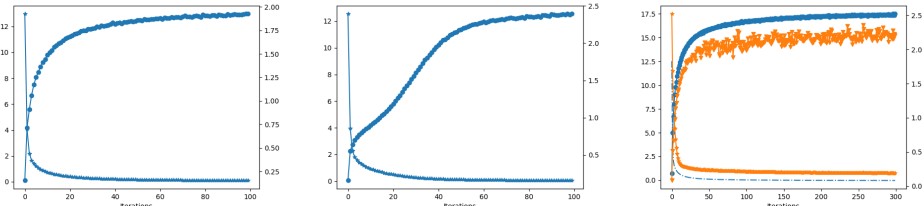

Figure 4: Training curves of the three algorithms for 78D Gaussian benchmark (the increasing line is the transport cost, and the decreasing line is the $D_{\mathrm{KL}}$. Left: the SL method; middle: the AL method; right: the ADMM method, orange line is for the first network and blue line is for the second network).

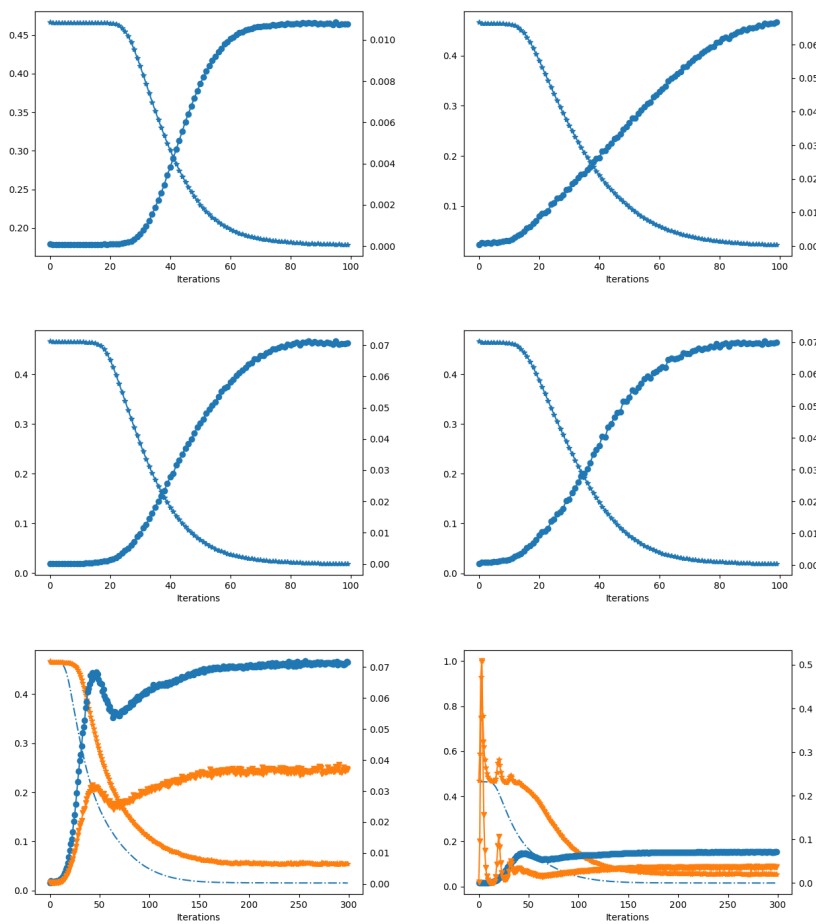

Figure 5: Graph of the training curves, increasing line for transport loss and decreasing line for the $D_{\mathrm{KL}}$. Top: SL with $\lambda = 0.1$(left), $\lambda = 10$(right) Middle: AL with $\lambda = 0.1$ (left), $\lambda = 10$ (right). Bottom: ADMM with $\Lambda = 0.1 \cdot \mathbf{1}$ (left) and $\Lambda = 10 \cdot \mathbf{1}$ (right).

For the WGAN example, we use a fully connected neural network with hidden layers of width 400. Gradient penalty is used for the WGAN-GP and for the SL, AL and ADMM methods, the alogrithm is described in section 3.4 and $\lambda = 1$ is used in the SL and $\rho = 10^{-4}$ is used for AL and $\rho = 10^{-5}$ is used for the ADMM.

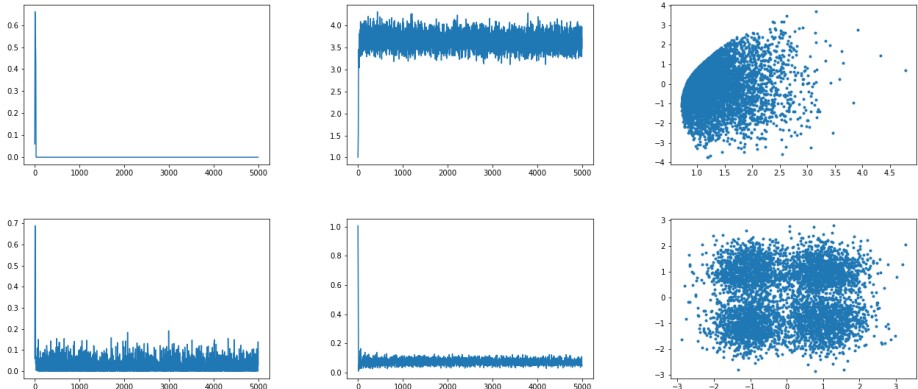

Figure 6: Graph of the results of WGAN without gradient penalty and the SL method; Left to the right: the discriminator loss, the generator loss and the generated sample; top: WGAN without gradient penaly, bottom: the SL method (equation (12))

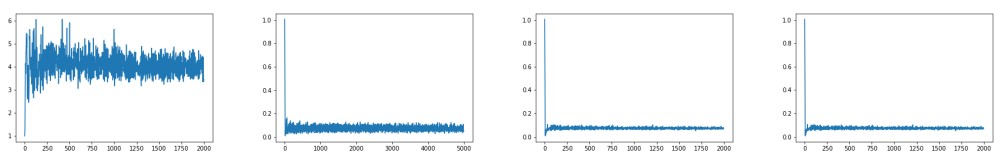

Figure 7: Graph of the transport distances. Top (left to right): WGAN-GP, SL, AL and ADMM

