# OpenReview forum: "OTCOP: Learning optimal transport maps via constraint optimizations"
_ICLR.cc/2023/Conference — Submitted to ICLR 2023_

### Official Review · Reviewer_rwsz · 2022-10-23

**Confidence:** 4
**Correctness:** 4
**Technical Novelty And Significance:** 3
**Empirical Novelty And Significance:** 3
**Recommendation:** 6

**Clarity, Quality, Novelty And Reproducibility:**

The idea of using constraint optimization to solve optimal transport problem has been existed for a well. By adding the penalty terms to the objective function with Lagrange multiplier is conventional. The ADMM algorithm is relatively novel than the other two methods. The theoretic proofs are convincing, especially theorem 3 guarantees the convergence of the proposed algorithm, this rigorous result has high practical value as well. But the scope of theorem 2 is limited, only for normal distributions. The paper is well written, all the mathematical formulations, key concepts, proofs are very clean and succinct. The algorithms are given in details and the experimental results are easy to reproduce.

**Details Of Ethics Concerns:**

This research is fundamental and mainly focuses on theoretical aspects.

**Strength And Weaknesses:**

The strengths of the current work
1. Propose novel algorithm to solve optimal transport map method directly using Monge's formulation with constraint optimization.
2. Three different optimization algorithms are formulated, evaluated and compared.
3. Theoretic proofs for the existence of the solution (Theorem 1 and Theorem 2), and the convergence of ADMM (Theorem 3)
The mathematical formulation and proofs are clean, rigorous and thorough.

The weakness of the current work is that the experiments are conducted on simple datasets (MNIST), it is desirable to test the theoretic results on real world datasets. Another weakness is that by using Yau and Gu's geometric variational algorithm, the problem of finding Monge Optimal transport map is reduced to a convex optimization problem, so the theoretic setup is easier and the optimization can be carried out using Newton's method. It is helpful to compare the proposed method with this convex optimization approach. Another weakness is that the proposed method can not find the singularity of the OT map precisely, since it use soft constraints (mollifiers) which will obscure the boundary of the supports.

**Summary Of The Paper:**

This work proposes a direct constraint optimization method to solve optimal transport maps using the original Monge's formulation. Three differential algorithms are studied:  the Langrangian multiplier method, the augmented Lagrangian method, and the alternating direction method of multipliers (ADMM). The proposed method improve the accuracy of learned optimal transport maps on high dimensional benchmarks, reduce the regularization effects and better learn the target distributions at a lower transport cost.

The contributions are
1. integrate three constraint optimization algorithms including the Standard Lagrangian (SL), the Augmented Lagrangian method (AL) and the Alternating Direction Method of Multipliers (ADMM) with neural networks to solve the Monge problem of optimal transport
with provable guarantees (Theorem 1-3).
2. show that the proposed  method is able to find an accurate optimal transport map between Gaussian distributions, both theoretically (Theorem 2) and experimentally. Moreover, the method is applied to WGAN and show that our method can find a generative map with lower transport cost while not sacrificing the quality of outputs.
3. Three algorithms are compared and it is found that  the SL algorithm introduces errors but is simple and easy to implement, while AL and ADMM algorithms can find exact results and are more robust, and ADMM gives a lower transport cost in general.

**Summary Of The Review:**

This work proposes a direct constraint optimization method to solve optimal transport maps using the original Monge's formulation. Three differential algorithms are studied:  the Langrangian multiplier method, the augmented Lagrangian method, and the alternating direction method of multipliers (ADMM). The proposed method improve the accuracy of learned optimal transport maps on high dimensional benchmarks, reduce the regularization effects and better learn the target distributions at a lower transport cost.

The paper is well written, the mathematical formulations are rigorous and elegant, the theoretic proofs are convincing. The preliminary numerical experiments support the claims. The work can be further improved : enlarge the scope of theorem 2, add more real world testing experiments. Furthermore, comparisons with other approaches which also solve the problem directly, such as the geometric variational method by Yau et al, which reduce the problem to convex optimization.

---

> ### Author Response · Authors · 2022-11-19
> **Response to Reviewer rwsz**
>
> Many thanks for reviewing our paper and providing the valuable comments. We thank you for appreciating the presentation of our content. We have uploaded a revised version of our paper. Below is a detailed response to your concern.
>
> 1. As the reviewer suggested, we are working on applying our methods to more complicated problems. Considering the theoretical results and successful application in simple datasets, we expect our methods to be applicable to more complicated problems.
>
> 2. Thanks for referring Yau and Gu's geometric variational algorithm to us. Since they transfer the problem into a convex one, the constraint optimization technique can also be applied and we expect better theoretical results. Different from our work, they use the Kantorovich duality in getting the optimization problem. We will work on this in the future.
>
> 3. Since we assume the distributions to be absolutely continuous singularity of the transport map might not be an issue theoretically. However, this may become an issue when applying our methods to more complicated problem. A preprocessing of the dataset may be needed before applying our methods, for example dequantization can make the discrete dataset continuous.
>
> 4. We have added a nonlinear version of applying the constraint techniques (use test function as multiplier) in the modified version, which better utilizes the nonlinearity nature of neural networks.

---

### Official Review · Reviewer_ZQmE · 2022-10-24

**Confidence:** 4
**Correctness:** 3
**Technical Novelty And Significance:** 3
**Empirical Novelty And Significance:** 3
**Recommendation:** 6

**Clarity, Quality, Novelty And Reproducibility:**

In this paper, the authors combine traditional numerical methods with deep learning methods so as to learn the optimal transmission on a high-dimensional benchmark. The theoretical derivation is relatively sound, but the relevant experiments are missing.

**Strength And Weaknesses:**

Strength:
(1) The article has sufficient theoretical derivations and proofs for the three algorithms.
Weaknesses:
(1)  Although it is illustrated that traditional numerical methods are not applicable to high-dimensional problems, relevant comparative experiments and graphs are missing.
(2) Lack of comparative experiments with existing learning-based methods.

**Summary Of The Paper:**

Combining traditional numerical methods for solving the Monge equation with deep learning, thus compensating for the limitations of numerical methods for high-dimensional data and the need for deep learning networks based on Kantorovich duality and complex network structures.

**Summary Of The Review:**

In general the article would be more valuable if some comparative experiments with traditional numerical algorithms were added, as well as some visual schematics.

---

> ### Author Response · Authors · 2022-11-19
> **Response to Reviewer ZQmE**
>
> Many thanks for reviewing our paper and providing the valuable comments.  We thank you for appreciating the presentation of our content. We have uploaded a revised version of our paper.
> Below is a detailed response to your concern.
>
> We have added some numerical experiments to demonstrate our algorithm. There are already some other work doing the comparison of different optimal transport solvers, for example "Korotin, Alexander, et al. "Do neural optimal transport solvers work? a continuous wasserstein-2 benchmark." Advances in Neural Information Processing Systems 34 (2021): 14593-14605." As for the traditional methods that use PDE to solve optimal transport problems, they work probably only under dimension 3, since in higher dimensions, one needs a grid for computation and the computation cost can be very high.
>
> As you suggested, we also plan to do more numerical experiments to see how our method work in more complex problems. Considering the theoretical results and our benchmark experiments, we expect the method to be also applicable to larger scale problems.

---

### Official Review · Reviewer_5X5o · 2022-10-24

**Confidence:** 3
**Clarity, Quality, Novelty And Reproducibility:** The paper is clearly written and the …
**Correctness:** 4
**Technical Novelty And Significance:** 3
**Empirical Novelty And Significance:** 2
**Recommendation:** 3

**Strength And Weaknesses:**

Strength:
- the paper is well-written
- the discussion on the Gaussian example is very helpful in understanding the proposed approach, its advantages and disadvantages

Weakness:
- the paper is motivated by shortcomings of the existing approaches in terms of special neural network structure or regularization, but the proposed approach seems to suffer from the same issues.
- the most challenging step of the approach is how to compute the penalty and perform the ADMM penalty step. It is not clear how to compute the KL divergence or any other type of distance (W1, IPM,..) in general. Minimizing the penalty term is the subject of all GAN papers and the way to do it exactly suffers from the issues that the paper is trying to avoid.


**Summary Of The Paper:**

The paper is considering the problem of learning the optimal transport maps with quadratic cost. In order to do so, they propose to apply a constrained optimization algorithm directly on the Monge formulation. In particular, the marginal constraint T#\mu = \nu is formulated as a penalty that involves a distance between T#\mu and \nu.  Then, constrained optimization algorithms, such as ADMM, are proposed to obtain the parameters of the map T. The proposed algorithm is explained and illustrated on Gaussian, mixture of Gaussians, and MNISST dataset.

**Summary Of The Review:**

Although it is a nicely written paper, the proposed approach suffers from the same issues that the paper is trying to avoid.

---

> ### Author Response · Authors · 2022-11-19
> **Response to Reviewer 5X5o**
>
> Many thanks for your reviewing our paper and that you find value in our work. We appreciate your valuable comments. We have modified the paper based on your suggestions. Below is a detail response to your concern.
>
> 1. We donot intend to solve the shortcomings of GANs. Our focus is on how to solve the optimal transport problem with a novel strategy.
>
> 2. We have added section 3.4 to describe how the KL divergence and Wasserstein distance are computed. KL divergence is computed using the network of normalizing flow. Moreover, inspired by your suggestion, we introduce a test function as Lagrangian multiplier and obtain a new way of regularization of GAN. Instead of use regularization on the discriminator network, we show in figure 6 that the transport cost regularization on the generator network also work and lead to a lower transport cost for the learned generative map.
>
> 3. Our methods can apply to different type of neural networks. Indeed, we use normalizing flow in our implementation with the KL divergence and use GAN for Wasserstein distance. One could use other distance function for probabilities and apply constraint optimization techniques on it.
>
> 4. We add Appendix 6 for the implementation details and provide the anonymous github link for the code.

---

### Official Review · Reviewer_3wmi · 2022-10-29

**Confidence:** 4
**Correctness:** 1
**Technical Novelty And Significance:** 2
**Empirical Novelty And Significance:** 1
**Recommendation:** 3

**Clarity, Quality, Novelty And Reproducibility:**

Clarify: The paper is generally not very clear and hard to follow. It makes a lot of hidden assumptions that need to be proven to be true. There is also no implement details, which make it difficult to evaluate its real performance and hard to reproduce.

**Strength And Weaknesses:**

Strength:
- It is interesting to solve the KL regularized OT problem. Combining them together should give us more understanding of the both problems.

Weakness:
- The method can only solve continuous OT problem supported on the whole Euclidean space, or it will encounter severe convergence problem since the KL divergence can easily be $\infty$.
- If the target distribution is discrete like MNIST, how to compute the KL divergence?
- For the Alg 1,2,3, it is unclear how to conduct the optimization, by sampling?
- From the current experimental results, either Fig. 4 or Fig. 5, it is hard to say that the proposed methods work well.
- No implement details.

**Summary Of The Paper:**

In this paper, the authors propose to directly solve the Monge's optimal transport problem under $L^2$ cost function, where the existence and uniqueness of the solution is guaranteed. Specifically, they use a neural network to parameterize the OT map, and try to solve the KL-divergence regularized OT problem proposed in equation (3). Furthermore, different Lagrangian multiplier methods are used to handle the constraints. Finally, the problems are solved by ADMM method. Some toy examples are used to illustrate the performance of the propose method.

**Summary Of The Review:**

Generally, the authors need to prove the hidden assumptions and do more experiments to show the performance.

---

> ### Author Response · Authors · 2022-11-19
> **Response to Reviewer 3wmi**
>
> Many thanks for carefully reading our manuscript and providing valuable comments. We have uploaded the modified version of our paper. Below is the detailed response to your concerns.
>
> 1. The KL divergence is not necessarily needed in our work, we can use any distance/divergence functional. We add section 3.4, illustrating how to apply the Wasserstein distance in our methods.
>
> 2. For discrete dataset such as MNIST, one can apply dequantization technique to transfer the problem to continuous space. For example, see "https://uvadlc-notebooks.readthedocs.io/en/latest/tutorial_notebooks/tutorial11/NF_image_modeling.htmlhttps://uvadlc-notebooks.readthedocs.io/en/latest/tutorial_notebooks/tutorial11/NF_image_modeling.html". To compute the KL divergence, we use a neural network of normalizing flow and the KL divergence can be computed, see section 3.4 of the updated manuscript for details.
>
> 3. For Alg. 1, 2, 3, the optimization is performed by sampling from the initial and target distributions. The optimization is done using gradient descent. We have add an anonymous github page for the code. We have added appendix A.6 for the implementation details and the github link.
>
> 4. We have modified the algorithm about GAN, see section 3.4 and figure 7 for details. Also the transport cost reduces significantly by using our methods while the target distribution is still learned. Please pay attention to the value of the axis since the range of plot varies from figure to figure.
>
> 5. The implement details are added to Appendix A. 6 with the anonymous  github link for the code.
>
> For the theoretical side of the paper, we do make some assumptions. For example, we assume all measures are absolutely continuous. While these assumptions may not be always realistic, they allow us to prove the convergence of the algorithms in certain limited situations.

---

> > ### Comment · Reviewer_3wmi · 2022-12-07
> > **Thanks for the response**
> >
> > I thank the authors for the response. After carefully reading the rebuttals and other reviewers' comments, I decide to keep my ratings.

---

> > > ### Author Response · Authors · 2022-12-08
> > > **Thanks**
> > >
> > > Thanks for taking the time in reading our work and providing the comments that help improve our work. We hope our rebuttal version address your concerns.

---

### Decision · Program_Chairs · 2023-01-20

**Decision:**

Reject

**Justification For Why Not Higher Score:**

* better description of the algorithm and the way to compute and handle the penalty.
* better empirical evaluations (other datasets and comparisons with other Monge map estimation approaches)

**Justification For Why Not Lower Score:**

N/A

**Metareview: Summary, Strengths And Weaknesses:**

The paper considers  a  constrained optimization method to solve the original Monge's formulation of OT.
Three different algorithms are considered, proposed and partially evaluated.

Most reviewers agree that the paper needs to be improved in several points (see reviewer's comment) and especially in
its experimental evaluation parts and its comparison with other approaches for learning Monge map.